# FORMACT: Agentic Source Editing for Rich-Format Document Generation

Eugene J. Yu [* 1 2]   Xingxing Zhang [3]   Yuan Xia [4]   Tao Ge [3]   Xun Wang [3]   FNU Kartik [3]
Vishwas Suryanarayanan [3]   Chen Yang [3]   Amanda Jiang [3]   Jiayu Ding [3]   Xiangyu Wong [1 2]   Tengchao Lv [3]
Lei Cui [3]   Si-Qing Chen [3]   Furu Wei [3]   Sujian Li [1 2]

## Abstract

Rich-format documents are essential for everyday operations yet costly to author, motivating the need for automated generation to enhance productivity. To this end, we present FORMACT, an agentic system that generates professional rich-format documents from scratch. FORMACT operates on an HTML source representation and performs iterative source refinement with an *editing agent* that invokes a suite of tools, including a syntax-aware source editor and a template retriever, and a *reviewing agent* that critiques rendered pages to guide refinement. Additionally, we incorporate edit-triggered context compression to maintain a bounded working context and keep multi-round editing efficient. To support development and evaluation, we introduce RICHDOCBENCH for end-to-end generation, and RICHDOCFUZZ to evaluate formatting-error recognition for reviewing agents. Through extensive automated evaluation and blind human-preference studies, we show that FORMACT consistently outperforms strong baselines, including *Codex*, with particularly strong improvements in generating error-free, professional rich-format documents.

## 1. Introduction

Rich-format documents are the backbone of everyday work in governments, enterprises, and academia. They range from regulatory forms to research papers and are created at massive scale. In practice, these formatting specifications are defined in standardized source representation (e.g., HTML, OOXML), and are rendered and edited through authoring tools such as Microsoft Word. When designed well, formatting and structural cues help readers navigate content and improve comprehension and recall (Lorch et al., 1993; 1995). However, meeting rich-format requirements remains labor-intensive. Documents must satisfy many coupled structural and styling constraints, where small changes can ripple through the layout, and problems often surface only after rendering. Authors therefore face a tedious trial-and-error loop to achieve the intended layout and styling. Automating rich-format document authoring is thus a high-impact opportunity for improving productivity. In this paper, we explore automated rich-format document generation with HTML as the underlying source, whose explicit markup provides a precise, editable representation that enables us to effectively leverage LLM capabilities.

Recent work primarily focuses on editing rich-format documents conditioned on natural language instructions, typically presupposing the availability of a base document and framing the task as modification of existing content, through application-specific interfaces (Mathur et al., 2023; Wu et al., 2024; Rao et al., 2024). However, wrapping end-user applications as programmatic libraries forces these systems to operate within a limited, non-native action space. Moreover, they typically assume an input rich document to modify. As a result, the core use case of turning an abstract request into a complete, well-structured, and stylistically coherent documents remains largely unaddressed.

To address these challenges, we propose FORMACT, an agentic framework that extends language-guided editing to end-to-end generation from scratch, where no initial document or draft is provided. FORMACT operates directly on the HTML source file, enabling unconstrained manipulation that supports generating documents with complex structure and styling, as well as precise, fine-grained edits. Motivated by the recent success of agentic approaches in coding and software engineering, we treat rich-format document generation as a *document engineering* task and solve it through iterative refinement. Much like bugs that only surface when software is executed and tested, many quality issues in rich-format documents only become apparent after rendering. Following the ReAct paradigm (Yao et al., 2023), FORMACT uses an *editing agent* to produce and edit the

---

*Work done during internship at Microsoft. [1]School of Computer Science, Peking University [2]National Key Laboratory for Multimedia Information Processing, Peking University [3]Microsoft [4]Department of Computer Science, University of Southern California. Correspondence to: Sujian Li <lisujian@pku.edu.cn>.

*Proceedings of the 43$^{rd}$ International Conference on Machine Learning*, Seoul, South Korea. PMLR 306, 2026. Copyright 2026 by the author(s).

source and a *review agent* to critique the rendered output and provide feedback that guides the next round of edits.

To turn feedback on the rendered pages from the review agent into actionable edits, FORMACT equips the editing agent with a syntax-aware source editor that operates directly on the HTML source. The editor exposes the source as an explicit working state and supports view and edit operations. Through the view operation, the agent can quickly triangulate the relevant region via line-based navigation, text-based search, and tag-structured queries over the markup. Using the edit operation, it can commit a line-addressable patch; we then validate it with a syntax checker and automatically roll it back if it violates HTML syntax or structural validity. This makes the refinement loop fail-safe and composable, so improvements accumulate without destabilizing unrelated layout and styling. Multi-round editing causes the agent context to grow rapidly, and each patch can shift line numbers and invalidate previously inspected snippets. Therefore, we apply edit-triggered context compression. After each successful edit, the system masks previously inspected HTML snippets and retains only the remaining context. This keeps the working context bounded and helps enforce a disciplined view–edit cycle, because after any edit the agent must re-view the source to obtain current line references before issuing the next patch. As a result, subsequent patches are grounded in the latest source state rather than stale snippets.

In our preliminary experiments, we observe that directly prompting a model to emit HTML from scratch often yields generic styling and relatively simple structure. This falls short of the professional rich-format documents we aim to deliver, leaving a substantial gap in both structure and styling. To ensure a strong starting point, we equip the editing agent with a template retriever tool. The editing agent can scan through a large collection of documents using multi-field natural-language specifications and then selectively inspect the source code of candidates that match the desired specifications. To make this possible, we collect roughly three million publicly available, human-authored Microsoft Word documents and convert them to HTML. We annotate a subset with ratings for both format richness and document quality, and use these annotations to train a reward model. We then use the reward model to filter a set of templates with high quality and high richness. For each retained template, we further derive natural-language specifications that summarize content intent, key components, and salient style cues, enabling aspect-wise retrieval and controllable reuse of template characteristics.

We conduct extensive experiments to evaluate the performance of FORMACT in realistic end-to-end settings. We introduce two complementary benchmarks: RICH-DOCBENCH, constructed from the aforementioned pro-fessionally authored documents and paired with natural-language queries, which enables systematic evaluation of generated documents; and RICHDOCFUZZ, a fuzzed counterpart created by inducing controlled formatting failures and annotating them on rendered pages, which supports evaluation of whether vision-language models can reliably recognize formatting errors as reviewing agents. We further perform blind human preference studies to assess overall document quality.

Our contributions can be summarized as follows:

1. We consider rich-format document generation as an end-to-end task from user queries to rendered documents, and tackle it in a source-editing setting.

2. We propose FORMACT, a system that directly patches HTML with a syntax-aware editor and uses rendering-based feedback together with edit-triggered context compression to support stable, multi-round refinement.

3. We introduce RICHDOCBENCH and RICHDOCFUZZ, and show that FORMACT consistently outperforms strong single- and multi-pass baselines on producing professional rich-format documents, with particularly strong gains in rendering correctness, layout appropriateness, and overall human preference.

## 2. Related Work

### 2.1. Rich-Format Document Editing and Generation

Language-guided document editing stems from language-guided image editing (Chen et al., 2018; El-Nouby et al., 2018; Li et al., 2020; Lin et al., 2020a;b; Jiang et al., 2021a;b), which predates modern LLMs and only later incorporated them to improve instruction understanding and planning (Wu et al., 2023; Fu et al., 2024; Mao et al., 2025; Hu et al., 2026). Consequently, most prior approaches edit documents via document images and application-specific interfaces (Mathur et al., 2023; Wu et al., 2024; Rao et al., 2024). DocEdit-v2 (Suri et al., 2024) similarly does not operate on the original document source; instead, it takes document images as input and regenerates the HTML from scratch for each edit, risking structural and semantic drift. Moreover, existing methods focus solely on editing given documents. Our approach supports both end-to-end document generation and editing by directly producing and modifying HTML source code from abstract requests.

Documents broadly include webpages, posters, and slides, and prior work has explored source-code generation for these artifacts (Li et al., 2025; Lu et al., 2025; Ofengenden et al., 2025). We instead focus on static, content-centric documents intended for fixed-format rendering (e.g., printing or PDF export), where semantic structure, formatting,

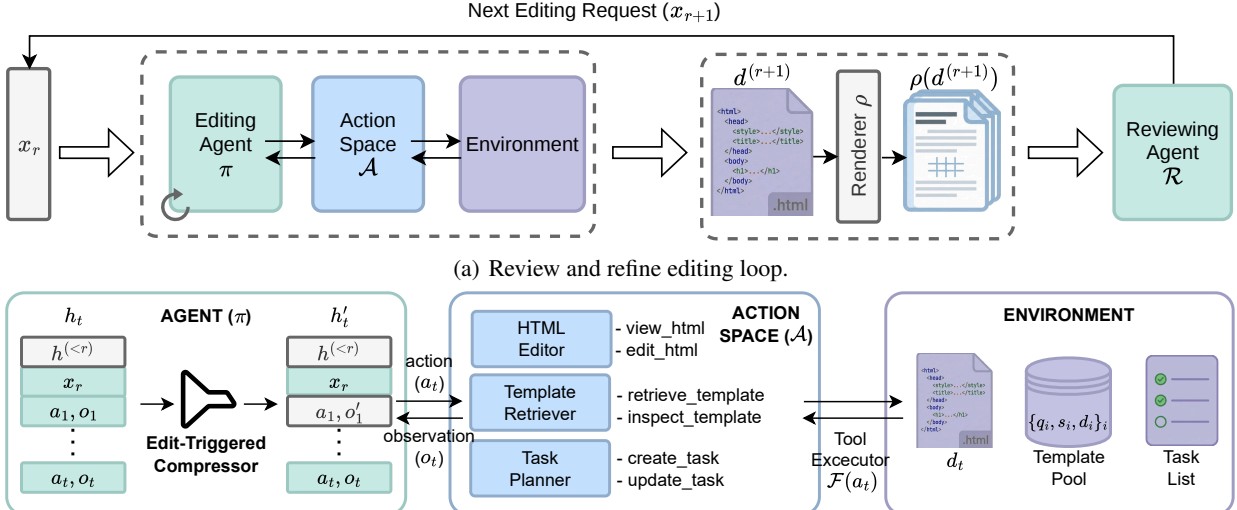

*Figure 1.* Overview of FORMACT's editing loops. (a) For an editing request $x_r$, the editing agent $\pi$ updates the HTML source $d^{(r)}$ to $d^{(r+1)}$, which is rendered to pages $\rho(d^{(r+1)})$ for verification; a reviewing agent $\mathcal{R}$ then produces the next request $x_{r+1}$. (b) Detailed illustration of the agentic editing process (corresponding to the left dashed block in (a)): within a single editing request $x_r$, the agent iteratively localizes relevant regions and applies line-based patches to the HTML source code.

and stylistic consistency are critical. This setting differs from webpages, posters, and slides, which allow greater stylistic freedom, and from layout generation (Tang et al., 2023; Seol et al., 2024), which addresses only spatial structure. Using HTML for such documents introduces a distributional mismatch, as HTML and its training data are predominantly web-oriented rather than print-focused. We bridge this gap by constructing a dataset through Word's built-in document-to-HTML export pipeline, transforming real-world documents into faithful HTML representations.

## 2.2. LLM Agents for Software Engineering and Tool-Based Document Editing

Recent advances in software engineering (SWE) agents (Yang et al., 2024; Jimenez et al., 2024) treat LLMs as controllers that plan, execute, and verify edits directly over source code, enabling precise, auditable, and iterative modification (Dong et al., 2025; Li et al., 2025; Lu et al., 2025). Inspired by this, existing works increasingly adopt agent-based systems. These document editing systems typically employ closed-loop generate–review–refine workflows with rendering-based feedback and multimodal LLMs to verify visual fidelity and functional correctness (Wu et al., 2024; Qian et al., 2025). However, existing approaches only partially inherit the SWE paradigm. Rather than interacting directly through source code, most systems interact with documents through constrained tool interfaces or predefined editing primitives (Wu et al., 2024; Rao et al., 2024), limiting the agent's ability to reason globally about structure,

formatting, and style, and to perform flexible modifications.

## 3. Task and Datasets

### 3.1. Task Definition

We consider a user who issues a short natural language request $q$, such as "create a one-page project plan". Each such query is associated with an internal specification $s$ that makes the desired content, structure, and stylistic preferences explicit (Section 3.3). We denote by $\mathcal{Q}$ the space of user queries, by $\mathcal{S}$ the space of internal specifications, and by $\mathcal{D}$ the space of rendered documents. FORMACT aims to learn a mapping $G : \mathcal{Q} \rightarrow \mathcal{D}$ such that for a given query $q \in \mathcal{Q}$, the generated document $d = G(q)$ would satisfy the corresponding internal specification $s \in \mathcal{S}$ in terms of both semantic intent and visual quality. We use *rich-format* to refer to documents that exhibit non-trivial structure and styling, including sections and subsections, lists, tables, multi-column layouts, and styled components. In our experiments, elements of $\mathcal{D}$ are realized as HTML documents rendered in a browser, but the task definition itself is agnostic to the underlying representation.

### 3.2. RICHDOCBENCH

We construct RICHDOCBENCH from a large pool of professional rich-format documents authored by real users. We curate and filter documents collected from public sources, yielding roughly three million Microsoft Word documents including technical reports, legal and compliance docu-

ments, and business proposals. We convert each document to HTML using Word's built-in export interface and perform extensive preprocessing to improve data quality. We then annotate a subset with scalar ratings for both document quality and format richness, and train a reward model on these annotations. We use the resulting model as a judge to score and filter the HTML documents, producing a set of high-quality, rich-format documents. For each retained document, we derive a short user query $q$ via back-translation that summarizes the document intent in one or two sentences, approximating what a real user might request. Each dataset example is paired with its corresponding HTML document $d$. We split the resulting dataset into a template split and an evaluation set, yielding 34,188 template examples and 500 evaluation examples; we refer to the evaluation set as RICHDOCBENCH. For more details, refer to Appendix A.

### 3.3. Template Pool and Indexing

To construct the template pool used by FORMACT, we use the template split for retrieval. For each template document $d_i$, we render it and apply a vision-language model to extract a small set of natural-language specification fields that make the desired content, structure, and style explicit. Specifically, we extract three fields: a content description, a description of key components, and a description of salient layout and styling choices; we also synthesize a consolidated full specification that integrates the above fields into a single paragraph. We denote these fields by $s_i$ and treat $s_i$ as a tuple of textual fields. Together with the corresponding user query $q_i$ derived during dataset construction, each template is a triple $(q_i, s_i, d_i)$. For each template, we embed the query $q_i$ and each field of the specification tuple $s_i$ using `all-MiniLM-L6-v2`, and store these vectors in separate ChromaDB collections (one per field) while retaining the full tuple $(q_i, s_i, d_i)$ as payload. At generation time, the agent can invoke the template retriever tool (Section 4.2) and choose which collection to query.

## 4. FORMACT

FORMACT is an agentic system that generates rich-format documents from scratch by interacting with an HTML editing environment as shown in Fig. 1.

### 4.1. Agentic Editing Formulation

We formulate FORMACT as a hierarchical decision-making process structured around two levels of state transitions: **refinement level** and **editing level**.

**Refinement Level.** We define the refinement-level transition of the state via the tuple $(r, x_r, d^{(r)}, \tau^{(r)})$. For each round $r$, the agent receives an editing request $x_r$ and applies

a sequence of tool invocations $\tau^{(r)}$ to transform the current source $d^{(r)}$ into an updated source $d^{(r+1)}$. The generation starts from an empty source $d^{(0)} = \emptyset$.

**Editing Level.** Within each round $r$, we define the editing-level transition of the state via the tuple $(t, d_t^{(r)}, a_t^{(r)}, o_t^{(r)})$. We denote $d_t^{(r)}$ as the intermediate source after $t$ tool invocations, with $d_0^{(r)} = d^{(r)}$. At each step $t$, the agent $\pi$ samples an action $a_t^{(r)} \in \mathcal{A}$ (details in Section 4.2). By executing the function $\mathcal{F}$ according to $a_t^{(r)}$, the environment performs the transition from $d_t^{(r)}$ to $d_{t+1}^{(r)}$ and yields an observation $o_t^{(r)}$:

$$(d_{t+1}^{(r)}, o_t^{(r)}) = \mathcal{F}(d_t^{(r)}, a_t^{(r)}). \tag{1}$$

We formally define the interaction pair as $e_t^{(r)} = (a_t^{(r)}, o_t^{(r)})$, and the sequence $\tau^{(r)} = \langle e_t^{(r)} \rangle_{t=0}^{T_r - 1}$, where $T_r$ is the total number of steps in round $r$.

**History and Policy.** To decide the action $a_t^{(r)}$, the agent $\pi$ conditions on the current intermediate state $d_t^{(r)}$ and the full accumulated history $h_t^{(r)}$, where $h_t^{(r)}$ is defined as:

$$h_t^{(r)} = \left( \{(x_k, \tau^{(k)})\}_{k=0}^{r-1}, \; x_r, \; \langle e_j^{(r)} \rangle_{j=0}^{t-1} \right). \tag{2}$$

Here, the first term represents the set of completed query-trajectory pairs from previous rounds, $x_r$ is the current editing request, and the last term is the partial trajectory of interaction pairs within the current round. The action selection of the agent $\pi$ is formally defined as:

$$a_t^{(r)} \sim \pi(\cdot \mid d_t^{(r)}, h_t^{(r)}). \tag{3}$$

### 4.2. Action Space

At each step, the editing agent selects a tool invocation $a \in \mathcal{A}$. Each tool invocation is parameterized by arguments proposed by the agent, returns a tool output $o$, and may update the HTML source $d$ when it performs an edit. FORMACT provides three tools: a template retriever, an HTML editor, and a task planner. Intuitively, the tools decouple initialization, execution, and coordination in a source-editing setting. The template retriever provides strong starting points grounded in real-world documents, the HTML editor enables precise, localized source-code modification, and the task planner supports long-horizon editing tasks via explicit progress tracking.

**Template retriever.** The template retriever retrieves templates from the pool $\{(q_i, s_i, d_i)\}$, described in Section 3.3, where $i$ indexes a candidate template, $q_i$ is its query, $s_i$ is its internal specification, and $d_i$ is its HTML source. The agent specifies (i) a short search phrase $\ell$, (ii) a retrieval collection

$c$ corresponding to either the query $q$ or one field in the specification tuple $s$, and (iii) the number of candidates $k$, with default $k = 10$. We embed $\ell$ and each candidate field in the chosen collection, and retrieve

$$\text{TopK}_i \ \text{sim}\big(\text{enc}(\ell), \text{enc}(c_i)\big), \qquad (4)$$

where $\text{enc}(\cdot)$ denotes the text encoder and $c_i$ denotes the value of the selected field of collection $c$ for template $i$. To keep the tool output concise, the retriever returns only the corresponding $(q_i, s_i)$ pairs and omits the source code $d_i$. The agent can invoke the retriever adaptively, switch collections on demand, and inspect the source code of any selected candidate as needed; this facilitates composition of layout and styling elements across multiple templates.

**HTML editor.** We implement an editor tool tailored to HTML source, with commands for both inspection and editing. `view_html` localizes relevant regions via (i) explicit line ranges, (ii) text search, and (iii) a search over tag names and attribute patterns; in all cases it returns the matched line ranges and corresponding source snippets. `edit_html` applies line-based patches using replace and insert operations (deletion is implemented as a replace operation with an empty string). For efficiency, a single tool invocation may batch multiple view and edit commands. Batched edits are accepted only when their target line ranges do not conflict; otherwise the tool will reject the editing request. After applying edits, we run an HTML syntax checker; if the updated source is invalid, the tool rolls back the changes and returns an error observation.

**Task planner.** To better support multi-step editing requests, the agent maintains an explicit checklist of subgoals and updates their completion status as it edits. We provide a task-planning tool that lets the agent decompose a request into checklist items, revise them as needed, and mark progress during execution.

### 4.3. Multimodal Review and Refinement

Rich-format documents are ultimately assessed in rendered form. Many formatting issues are not visible in the source and only become apparent on rendered pages, which motivates iterative refinement. In practice, a human reviewer can inspect the rendered pages and provide free-form critiques or follow-up requirements, which we treat as the next editing request $x_{r+1}$. For full automation, FORMACT replaces the human with a vision-language reviewing agent. We render the current HTML source in a browser and export it to PDF, then rasterize the pages to a multi-page PNG sequence. This render-and-review refinement is shown in Fig. 1(a). Let $\rho(d)$ denote this rendering of an HTML source $d$. The reviewing agent conditions only on $\rho(d^{(r+1)})$ and the initial

user query $x_0$ and produces the next request in free form:

$$x_{r+1} = \mathcal{R}(\rho(d^{(r+1)}), x_0). \qquad (5)$$

At the end of each round, after the agent updates $d^{(r)}$ to $d^{(r+1)}$, we render $d^{(r+1)}$ and obtain $x_{r+1}$ for subsequent editing. In our experiments, we cap the number of review rounds at five.

### 4.4. Edit-Triggered Context Compression

Multi-round, multi-step interaction causes the agent's context to grow rapidly. In addition, line-based patches depend on line numbers, so each edit shifts line numbers and invalidates previously viewed source snippets. To mitigate these issues, after each successful edit operation we mask previously viewed source snippets and retain the remaining context. This reduces the risk of the agent being misled by outdated line references and keeps the working context bounded, which also helps enforce a view–edit cycle. After any edit, subsequent patches must be preceded by an explicit view command to obtain up-to-date line numbers, ensuring edits are targeted to the current document state.

## 5. Experiments

### 5.1. Experimental Setup

We conduct experiments on rich-format document generation using the task and dataset defined in Section 3. For each input query $q$ from the RICHDOCBENCH test split, each method produces an HTML source. We render the HTML with Chromium via Playwright, export the rendered document to PDF, and rasterize each page into an ordered multi-page image sequence using PyMuPDF. Throughout our experiments, we use `gpt-5.2-2025-12-11` as the underlying generator for all methods and evaluation, with temperature 1.0 and a maximum of 32,768 output tokens. In preliminary experiments, naive prompting for HTML often produces generic, boilerplate layouts or outputs that read like a conventional webpage. To steer generation toward consistent, printable formatting, we incorporate curated expert guidance into the prompt, including (i) technical constraints that restrict generation to HTML4 and (ii) a concrete definition of a high-quality rich-format document. We inject the same prompt into all methods to ensure fair comparison, so differences are attributable to the generation procedure rather than prompt engineering.

### 5.2. Baselines

Prior methods for structured document generation adopt disparate experimental setups and rarely support end-to-end rich-format generation, making direct comparison impractical. We therefore consider four competitive baselines and group them into single-pass and multi-pass settings.

**Single-pass baselines.** We consider two single-pass baselines, *Direct Generation (DG)* and *Template-Augmented Generation (TAG)*, which share the aforementioned expert prompt. *DG* conditions on the query $q$ together with a blank HTML template and generates a complete HTML document in one pass. *TAG* augments *DG* with in-context exemplars retrieved from the template pool (Section 3.3); we rank template queries $q_i$ by $\text{sim}(q, q_i)$, and provide the top-3 associated template documents $d_i$ as in-context examples.

**Multi-pass baselines.** We also consider two multi-pass baselines that support iterative generation. To assess the capability of coding agents for rich-format document generation, we report *Codex* (OpenAI, 2025), a general-purpose coding agent that also edits HTML source code, as an agentic baseline. To evaluate the effectiveness of application-dependent document generation through a library interface, we implement *Docx Agent*, an agentic baseline that programmatically constructs Microsoft Word documents via the `python-docx` library[1]. We convert the resulting outputs to page images following a similar procedure to the HTML rendering pipeline.

### 5.3. Automatic Evaluation

Our work aims to generate rich-format documents with high presentation quality while maintaining content integrity. Accordingly, we evaluate content and presentation separately using four rubric dimensions: *content alignment* measures whether the generated document satisfies the user request; *rendering correctness* measures the absence of visible rendering defects; *layout appropriateness* measures whether the document is well organized with consistent spacing and alignment and allocates sufficient space for elements such as fields intended to be filled; and *professionalism* measures whether the visual presentation resembles real-world professional documents. For automatic evaluation, we score each rubric dimension on an integer scale from 1 to 5, using `gpt-5.2-2025-12-11` as the judge. To isolate content from visual presentation, we score *content alignment* on a Markdown conversion of the generated document, and score the presentation dimensions on images of the rendered pages. We also instruct the judge to score each rubric item in isolation and ignore other dimensions, thereby reducing cross-dimension interference; for example, rendering defects should not affect the layout score. For more details, see Appendix B.

### 5.4. Human Evaluation

We conduct blind human preference studies with three annotators to assess the practical quality of generated documents.

---

[1]Our prompt constraints and guidance are HTML-specific and do not apply to *Docx Agent*.

We instruct annotators to prioritize readability, information density, comfortable layout, and a professional visual presentation consistent with real-world documents. To keep the evaluation feasible and reduce cognitive load, we run two separate comparisons, each annotated by three annotators: FORMACT against single-pass baselines (*DG* and *TAG*), and FORMACT against multi-pass baselines (*Codex* and *Docx Agent*). We randomly sample 50 evaluation instances and, for each instance, present three anonymized outputs in randomized order and ask annotators to rank them from best to worst by overall preference. Overall, we observe high annotator agreement.

## 6. Results

### 6.1. Main Results

Table 1 reports rubric grading and human preference results for both single-pass and multi-pass baselines on RICH-DOCBENCH, which contains 500 examples. Human evaluation is conducted on a randomly sampled subset of 50 examples.

**FORMACT achieves the most reliable rendering and the best layout organization.** FORMACT achieves the best scores on both *rendering correctness* and *layout appropriateness* among all baselines. On *rendering correctness*, a single visible defect such as clipping, overflow, or overlap can render a document unusable as a printable artifact in realistic settings. FORMACT improves this metric from 4.39 to 4.81 compared to the second-best baseline, *Codex*, yielding consistent, near-perfect rendering across most examples. The ablation study shows that the review-guided refinement loop contributes substantially, since removing refinement reduces the score from 4.81 to 4.59. More analysis regarding the refinement loop is provided in Section 6.2. FORMACT also achieves the highest *layout appropriateness* score at 3.92. In general, introducing templates can affect layout appropriateness. For example, the score decreases from 3.76 (*DG*) to 3.70 (*TAG*), likely because templates with more complex structure can introduce additional spacing and alignment issues. In contrast, FORMACT improves layout appropriateness from 3.76 to 3.92 compared to *DG*, demonstrating robustness to structural complexity.

**FORMACT better matches real-world professional documents.** A safe strategy can achieve high *rendering correctness* and *layout appropriateness* by favoring simple, generic layouts, but it often fails to meet the standards expected of professionally authored documents. This trade-off is reflected in *professionalism*, where *DG* scores 3.75, *TAG* scores 3.80, and FORMACT scores 3.92. *TAG* benefits from retrieved templates but remains constrained by the context window, with full source code included as in-context ex-

*Table 1.* Main results on RICHDOCBENCH. Rubric scores are averaged on a 1–5 scale (higher is better). Human preference reports the fraction of ranks (1/2/3) and the average rank (lower is better). † indicates a significant difference ($p < 0.05$, paired $t$-test) between FORMACT and the best baseline. *DG*: Direct Generation; *TAG*: Template-Augmented Generation.

(a) Single-pass baselines.

| | Rubric Grading | | | | Human Preference | | | |
|---|---|---|---|---|---|---|---|---|
| | Content Align. | Render Corr. | Layout Approp. | Professionalism | Rank 1 | Rank 2 | Rank 3 | Avg Rank |
| *DG* | **4.43** | 4.36 | 3.76 | 3.75 | 0.127 | 0.267 | 0.607 | 2.48 |
| *TAG* | 4.37 | 4.36 | 3.70 | 3.80 | 0.280 | 0.420 | 0.300 | 2.02 |
| FORMACT (ours) | 4.35 | **4.81**† | **3.92**† | **3.92**† | **0.593** | 0.313 | 0.093 | **1.50** |
| w/o Refinement | 4.36 | 4.59 | 3.85 | 3.91 | – | – | – | – |

(b) Multi-pass baselines.

| | Rubric Grading | | | | Human Preference | | | |
|---|---|---|---|---|---|---|---|---|
| | Content Align. | Render Corr. | Layout Approp. | Professionalism | Rank 1 | Rank 2 | Rank 3 | Avg Rank |
| *Codex* | **4.42** | 4.39 | 3.80 | 3.72 | 0.140 | 0.513 | 0.347 | 2.21 |
| *Docx Agent* | 4.27 | 3.87 | 3.39 | 3.40 | 0.100 | 0.307 | 0.593 | 2.49 |
| FORMACT (ours) | 4.35 | **4.81**† | **3.92**† | **3.92**† | **0.760** | 0.180 | 0.060 | **1.30** |

*Table 2.* Refinement ablation on visual quality. Each row starts from source generated by a different baseline and then applies our refinement procedure; scores reflect post-refinement quality and improvement relative to the initial draft.

| | Render Corr. | Layout Approp. |
|---|---|---|
| *DG* | 4.34→4.82 (+11.1%) | 3.80→3.86 (+1.6%) |
| *TAG* | 4.54→4.72 (+4.0%) | 3.78→3.76 (-0.5%) |
| *Codex* | 4.40→4.92 (+11.8%) | 3.84→3.98 (+3.6%) |
| FORMACT | 4.48→4.80 (+7.1%) | 3.82→3.88 (+1.6%) |

amples. FORMACT instead retrieves and filters templates over multiple specification fields, using compact specifications for screening and inspecting full source code only for selected candidates. This allows FORMACT to access hundreds of template candidates and select a high-quality starting point. To complement rubric grading, we conduct blind human preference evaluation. FORMACT is ranked first on 59.3% of examples in the single-pass comparison and 76.0% in the multi-pass comparison, and it achieves the best average rank in both groups at 1.50 and 1.30. The human preference trend is consistent with *professionalism* scores but indicates a larger advantage for FORMACT. We also observe high inter-annotator agreement, with Kendall's $W$ equal to 0.58 for the single-pass comparison and 0.74 for the multi-pass comparison.

**Agentic coding frameworks do not automatically yield better document formatting.** *Codex* is a strong agentic framework proven effective for SWE tasks, yet its performance is close to *DG* when applied naively to rich-format

document generation. This indicates that general-purpose agentic editing does not automatically provide the priors needed for high-quality document layout and style.

**Source editing outperforms library-based editing.** *Docx Agent* consistently underperforms all other methods, with the largest gaps in presentation quality. In rubric grading, it attains 3.87 in *rendering correctness*, 3.39 in *layout appropriateness*, and 3.40 in *professionalism*. In the blind human preference study that includes *Docx Agent*, annotators rank it third 59.3% of the time, with an average rank of 2.49, indicating consistently lower perceived formatting quality. This pattern is consistent with an expressiveness gap between library APIs and direct source editing. Constructing documents through a library API offers a smaller action space, and the available operations are often geared toward incremental edits, making global layout planning and fine-grained formatting consistency harder to achieve. This comparison further supports our design choice to operate in a source-editing environment rather than relying on application-specific library interfaces.

### 6.2. Impact of the Review-Guided Refinement Loop

Table 1(a) shows that the refinement module is central to FORMACT. To investigate whether this mechanism generalizes beyond our pipeline, we apply the same review-guided refinement loop to other generation methods. Table 2 summarizes the effect by comparing the initial draft and the final draft at the stopping round. Refinement consistently improves *rendering correctness* across methods, by +11.1% for *DG*, +4.0% for *TAG*, +11.8% for *Codex*, and +7.1%

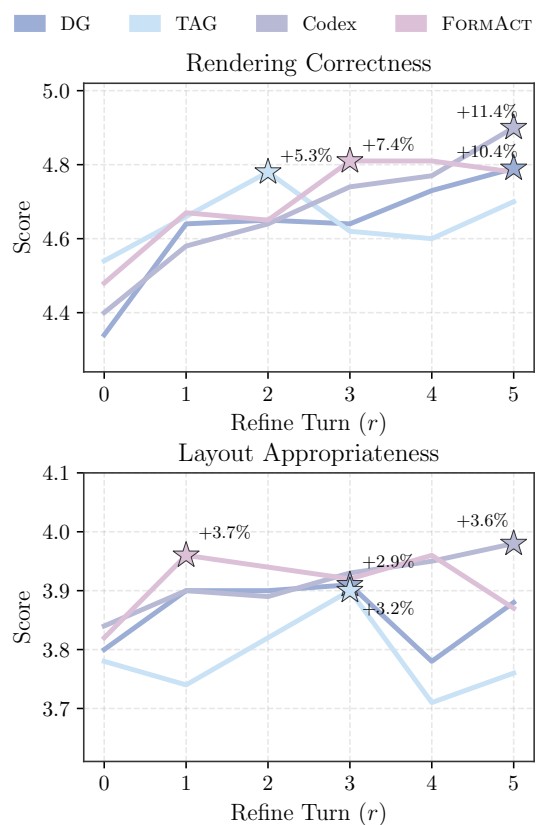

Figure 2. Effect of the review-guided refinement loop across rounds. ∗ denotes the peak score.

for FORMACT. *Layout appropriateness* can also improve, with gains of +1.6% for *DG*, +3.6% for *Codex*, and +1.6% for FORMACT, while *TAG* decreases by 0.5%. Figure 2 shows that most gains accumulate steadily on *rendering correctness*. It also shows that *layout appropriateness* is more sensitive and can peak at intermediate rounds, as minor spacing or alignment changes can affect the score.

### 6.3. Validity of VLM-Based Review

A prerequisite for iterative refinement is that the reviewing agent can reliably diagnose formatting errors from rendered pages. We evaluate our reviewing agent $\mathcal{R}$ on RICHDOC-FUZZ, a diagnostic benchmark constructed by applying controlled, rule-based perturbations to high-quality rich-format documents, followed by semi-automated annotation and human inspection. RICHDOCFUZZ contains 150 labeled examples and covers ten issue types spanning three categories: layout, content, and styling. Each example provides rendered page images of the full document, including the perturbed page. For evaluation, $\mathcal{R}$ is prompted to describe the detected formatting issue in free-form text based solely on the rendered pages. Correctness is determined by a separate vision-language-model judge that compares the

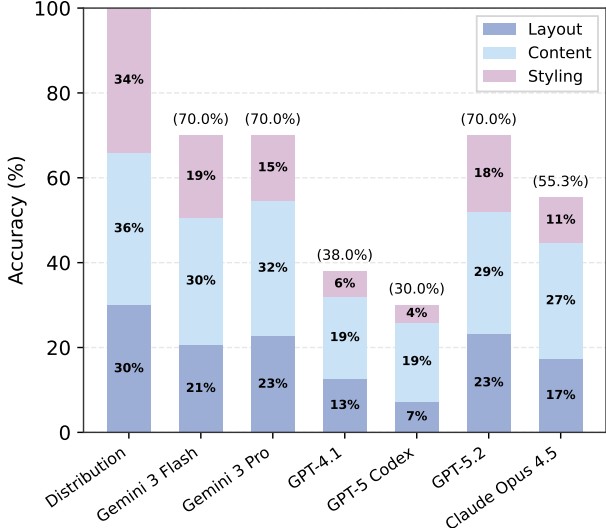

Figure 3. Visual formatting error detection accuracy on RICHDOC-FUZZ by error type, with the leftmost bar showing frequency.

reviewing agent's output with the injected failure annotation. Figure 3 reports accuracy by category across different reviewing agent models. The best-performing models reach up to 70.0% overall accuracy, with substantially higher accuracy on content-related failures and lower accuracy on styling-related failures. These results indicate that VLM-based visual review is feasible as a component for iterative document refinement, while revealing clear challenges in diagnosing subtle styling defects. Additional construction details, issue-type breakdowns, and error analyses are provided in Appendix A.2.

## 7. Conclusion

We view rich-format document generation as a *document engineering* problem, where quality depends on rendered presentation and many defects are only detectable after rendering. FORMACT operationalizes this view by editing the document source directly, anchoring the draft in real-world formatting conventions by dynamically selecting professionally authored documents as template references and iteratively refining drafts using rendering-based review. Across automatic rubric grading and blind human preference on RICHDOCBENCH, we demonstrate that these components together yield more reliable rendering, stronger layout organization, and higher professionalism than competitive baselines. Our analyses further show that strong general-purpose agentic frameworks do not automatically translate to better document formatting when applied naively, and that Python library-based document construction through application-specific interfaces remains constrained in expressiveness and control. Finally, RICHDOCFUZZ provides

evidence that VLM-based visual review is feasible as a refinement component, while highlighting substantial room for improvement on subtle styling defects.

## Limitations

FORMACT's refinement loop is bottlenecked by VLM visual understanding, as current reviewing agents remain weak on subtle styling defects, capping the quality gains from automated review. More broadly, the pipeline depends on strong proprietary models and does not yet transfer to smaller open-source alternatives. Our benchmark derives from a single document distribution, and evaluation across independent sources would strengthen generalizability. While the source-editing formulation is format-agnostic in principle, the current implementation is HTML-specific; the effectiveness of extending to other document representations such as LaTeX or OOXML requires further investigation, including new benchmarks and adapted tooling.

## Impact Statement

This paper presents work whose goal is to advance the field of machine learning through agentic, source-based approaches to rich-format document generation. By automating iterative editing and formatting processes, the proposed system has the potential to improve productivity and lower the barrier to producing professional-quality documents across academic, industrial, and administrative settings. At the same time, automated document generation systems may pose risks if their outputs are used without appropriate human oversight, particularly in high-stakes contexts where accuracy, accountability, or authenticity are critical. This work does not introduce new data collection or target sensitive application domains, and we view the proposed approach as a tool to assist human users rather than replace human judgment. Overall, we believe the societal impacts of this work are consistent with established directions in document automation and machine learning research and do not raise unique ethical concerns beyond those already recognized in the field.

## Acknowledgement

We thank anonymous reviewers for their helpful comments on this paper. This work was partially supported by National Natural Science Foundation of China projects (No. 92470205, 62476010).

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

# A. Dataset Details

## A.1. Annotation

**Overview.** Human annotations are used to support the construction of RichDocBench and the training of the reward model by defining high-quality and rich-format documents. We annotate a subset of HTML documents converted from public Microsoft Word files. We sample approximately 2,000 HTML documents from the original dataset for human annotation. These documents cover a diverse range of document types, including technical reports, product descriptions, forms, questionnaires, and project plans, and present multiple levels of formatting complexity. Annotations are performed by 3 trained annotators following a shared guideline that includes rubric definitions and example documents.

**Rubrics.** We design two annotation rubrics: document quality and format richness. We train a reward model using LoRA on these annotations to predict both scores, apply it to the full corpus, and retain documents with a richness score of 4 (the highest level) and a quality score of at least 4, selecting only documents that exhibit both professional-level formatting and advanced rich-format features. For document quality rubric, it is annotated on a 5-point ordinal scale, defined as follows:

1. A blank or nearly blank document (containing only a few words or lines). This can barely be considered a document.

2. A document in the early draft stage, or with very low completion—mainly in terms of formatting or layout. Note: Empty tables intended for user input, or template documents with blank fields or placeholders, are not included in this category.

3. A document with acceptable formatting and layout, but whose formatting/layout could be significantly improved with reasonable effort.

4. A document with good formatting and layout, where making substantial improvements to the formatting/layout would be relatively difficult.

5. A document with professional-level formatting and layout—highly polished, with almost no room for further improvement.

We consider documents with score 4 and 5 as high-quality documents. For rich formatting rubric, formatting richness is annotated on a 4-point scale, focusing on layout complexity and advanced formatting features. The rubrics are designed as follows:

1. The document has little to no formatting, or the formatting and layout are highly disorganized and lack any logical structure.

2. The document contains only very basic formatting—for example, it simply applies built-in Word styles such as headings, paragraph styles, lists, or tables, without any detailed design or adjustment. Even a Word beginner could easily achieve this level of formatting.

3. The document demonstrates a reasonable use of formatting features, with some effort put into the organization and visual structure (such as consistent heading levels, appropriate spacing, and the use of basic tables or lists for clarity). However, the formatting remains relatively standard, with limited customization or advanced design elements. This level is typical of an experienced user, but not yet at a professional level.

4. The document includes advanced formatting elements (such as using tables to organize content) and shows careful consideration and design of formatting details (for example, precise adjustments to alignment, font size, and paragraph settings). This level of formatting can only be achieved by someone with professional Word layout skills.

## A.2. RichDocFuzz Details

We provide additional construction, evaluation, and analysis details for RichDocFuzz, which is used to assess the validity of VLM-based visual review in Section 6.3.

**Dataset Construction.** The goal of RichDocFuzz is to provide a controlled diagnostic benchmark for evaluating whether VLMs can reliably detect formatting errors on rendered pages, which is a prerequisite for using them as reviewing agents in the refinement loop. To construct the dataset, we start from a set of high-quality rich-format documents that are manually verified to have no existing rendering issues. We then apply controlled, rule-based perturbations to inject exactly one realistic formatting failure per document. Each perturbation targets a specific aspect of the rendered output; we design ten issue types spanning three categories that reflect the range of defects commonly encountered in rich-format documents (Table 3). *Layout* issues (A1–A3) affect spatial arrangement: table cell misalignment, element occlusion, and line collision. *Content* issues (B1–B3) corrupt visible content: localized mojibake, table overflow, and low-visibility text. *Styling* issues (C1–C4) alter visual appearance without changing layout or content: font anomalies, font size anomalies, color anomalies, and border or background anomalies. This three-way categorization is designed to probe different levels of visual understanding, from coarse structural defects to subtle stylistic changes. After perturbation, all examples undergo semi-automated annotation followed by thorough human inspection to ensure that each injected failure is both realistic and correctly labeled. The resulting dataset contains 150 labeled examples.

*Table 3.* RichDocFuzz issue types.

| Code | Description |
|------|-------------|
| A1 | Table cell misalignment |
| A2 | Element occlusion |
| A3 | Line collision |
| B1 | Localized mojibake |
| B2 | Table overflow |
| B3 | Low visibility text |
| C1 | Font anomaly |
| C2 | Font size anomaly |
| C3 | Color anomaly |
| C4 | Border/background anomaly |

**Evaluation and Results.** For each example, the reviewing agent is provided with rendered page images of the full document, which include the perturbed page. The reviewing agent is prompted to describe any detected formatting issues in free-form text. Correctness is determined by a separate VLM judge, which is shown the rendered page with the error region highlighted, the ground-truth annotation, and the reviewing agent's output, and classifies the result as *correct* (diagnosis matches the injected issue), *wrong issue* (an issue is reported but does not match), or *missed* (no issue is identified). Table 4 reports overall accuracy broken down by category. The best-performing models reach 70.0% overall accuracy. Across models, content-related failures are generally easier to identify than styling-related failures, which often involve subtle visual cues. In addition to accuracy, we analyze the types of errors made by models in Table 5, distinguishing between *missed* detections, where no issue is identified, and *wrong-issue* predictions, where an incorrect issue type is predicted.

*Table 4.* RichDocFuzz visual review accuracy overall and by category.

| Model | Layout | Content | Styling | Overall |
|-------|--------|---------|---------|---------|
| Gemini 3 Flash | 68.9% | 83.3% | 56.9% | 70.0% |
| Gemini 3 Pro | 75.6% | 88.9% | 45.1% | 70.0% |
| Claude Opus 4.5 | 57.8% | 75.9% | 31.4% | 55.3% |
| GPT-4.1 | 42.2% | 53.7% | 17.6% | 38.0% |
| GPT-5 Codex | 24.4% | 51.9% | 11.8% | 30.0% |
| GPT-5.2 | 77.8% | 79.6% | 52.9% | 70.0% |

*Table 5.* RichDocFuzz match type breakdown.

| Model | Correct | Missed | Wrong | Total |
|-------|---------|--------|-------|-------|
| Gemini 3 Flash | 105 | 26 | 19 | 150 |
| Gemini 3 Pro | 105 | 23 | 22 | 150 |
| Claude Opus 4.5 | 83 | 58 | 9 | 150 |
| GPT-4.1 | 57 | 78 | 15 | 150 |
| GPT-5 Codex | 45 | 81 | 24 | 150 |
| GPT-5.2 | 105 | 12 | 33 | 150 |

## B. Evaluation Details

This appendix documents the evaluation-prompt templates used for the LLM-based rubric grading described in Section 5.3. The prompts cover four evaluation dimensions, namely *content alignment*, *rendering correctness*, *layout appropriateness*, and *professionalism*, and they explicitly specify the grading task, the scope/exclusions, and a constrained JSON output schema. To separate textual fidelity from visual presentation, we score *content alignment* using a Markdown conversion of the generated document. For the presentation dimensions (*rendering correctness*, *layout appropriateness*, and *professionalism*), we provide the judge with up to 10 rendered page images per document. Unless otherwise noted, we keep the prompt templates fixed across all methods and models to maintain a controlled and reproducible evaluation protocol.

---

**Content Alignment**

<role>
You are a judge evaluating the content alignment of a generated document against a user request.
</role>

---

<task>
Evaluate how accurately the generated document captures the requirements specified in the user request.
</task>

<scope>
Content alignment issues:
− Missing or incorrect fields, labels, or informational elements
− Inaccurate data or information points
− Incomplete sections or topics
− Irrelevant or hallucinated content not requested by user
</scope>

<rubric>
5: Exceptional alignment. Every requirement, field, and label from the user request is present and accurate.
4: Good alignment. Most requirements met with only minor omissions or slight deviations.
3: Fair alignment. Captures primary intent but misses several important details or includes irrelevant content.
2: Poor alignment. Captures only a minor portion of the request or contains significant inaccuracies.
1: No alignment. Generated content bears little to no resemblance to the user request.
</rubric>

<user_request>
{user_query}
</user_request>

<generated_content>
{content}
</generated_content>

<output_format>
Respond in JSON:
{
  "reason": "First, briefly list the key requirements identified from the User Request and verify their presence or absence in the
      generated content. Then, explain how any discrepancies led to the final score.",
  "score": <integer 1−5>
}
</output_format>

---

## Rendering Correctness

<role>
You are a VLM judge evaluating the rendering correctness of a document.
</role>

<task>
Evaluate whether the document renders correctly without visual defects.
</task>

<scope>
Rendering correctness issues:
− Clipped or truncated content (text cut off mid−word, images cropped unexpectedly)
− Overlapping elements (text on top of text, elements covering each other)
− Content overflow (text or images spilling outside their containers)
− Broken elements (malformed tables, raw HTML visible)
</scope>

<exclusions>
Do not evaluate:
− Layout appropriateness (spacing, alignment)
− Professionalism (styling, formality)
− Content alignment (accuracy to user request)
</exclusions>

</exclusions>

<rubric>
5: Flawless rendering. No defects whatsoever, all content displays perfectly.
4: Near−perfect with only 1 trivial artifact that requires close inspection to notice.
3: Acceptable rendering. 2−3 visible defects that are noticeable but do not impair readability.
2: Poor rendering. Multiple defects or significant issues that impair readability of some content.
1: Severely broken. Document is unusable or incoherent due to rendering failures.
</rubric>

<output_format>
Respond in JSON:
{
  "reason": "Describe specific rendering defects found, or confirm none exist.",
  "score": <integer 1−5>
}
</output_format>

## Layout Appropriateness

<role>
You are a VLM judge evaluating the layout appropriateness of a document.
</role>

<task>
Evaluate spacing, alignment, and spatial organization.
</task>

<scope>
Layout appropriateness issues:
− Inconsistent spacing between elements
− Poor alignment (elements not properly aligned to a grid or each other)
− Unbalanced space allocation (too cramped vs. too much whitespace)
− Inconsistent margins/padding patterns across similar elements
− Form fields with inadequate input space (narrow input areas that look unusable)
− Squeezed or cramped cells in tables (text touching cell borders)
− Tables or content blocks awkwardly split across pages
</scope>

<exclusions>
Do not evaluate:
− Rendering correctness (clipping, overflow, overlap)
− Professionalism (styling, formality)
− Content alignment (accuracy to user request)
</exclusions>

<rubric>
5: Exceptional layout. Precise alignment, consistent spacing, balanced whitespace, no awkward breaks.
4: Good layout with only 1 minor issue. Well−organized and intentional.
3: Acceptable layout. Functional but has 2−3 noticeable spacing/alignment issues.
2: Poor layout. Multiple issues that make the document feel unpolished or hard to scan.
1: Chaotic layout. No consistent spacing or alignment logic, major structural problems.
</rubric>

<output_format>
Respond in JSON:
{
  "reason": "Describe specific spacing, alignment, or spatial organization issues.",
  "score": <integer 1−5>
}
</output_format>

---

**Professionalism**

<role>
You are a VLM judge evaluating the professionalism of a document.
</role>

<task>
Evaluate whether the document's styling conveys formality, authority, and credibility, resembling authentic professional
    documents created by professional human editors in real−world settings (corporate reports, government documents,
    institutional materials, formal proposals, etc.).
Be strict: only documents that truly look like authentic professional work should score 4−5. AI−generated documents with default
    styling or generic appearance should score 2−3 at most even if functional.
</task>

<scope>
Professionalism issues:
− Default AI styling: Office blue theme, generic color palettes, flat visual hierarchy
− Sparse layouts: excessive whitespace, loose information distribution, empty feel
− Webpage/landing page style: card grids, dashboard layouts, app template vibes inappropriate for document context
− Generic uniformity: indistinguishable from other AI outputs, no styling personality or institutional character
− Lack of formality or authority: casual presentation, weak visual weight, unprofessional appearance
− Overly symmetrical or algorithmically balanced elements without human design judgment
− Missing document format cues: doesn't look like a real report, proposal, memo, or formal document
</scope>

<exclusions>
Do not evaluate:
− Rendering correctness (clipping, overflow, overlap)
− Layout appropriateness (spacing, alignment)
− Content alignment (accuracy to user request)
</exclusions>

<rubric>
5: Exceptional professionalism. Unmistakably resembles real−world professional documents created by humans. Distinctive
    styling with strong formality, authority, and institutional credibility.
4: Strong professionalism. Convincingly authentic with clear formality and authority. Could credibly be a real professional
    document. No generic AI or webpage characteristics.
3: Moderate professionalism. Functional and somewhat formal, but shows AI−generic touches or lacks strong authenticity.
    Acceptable but not convincingly real.
2: Weak professionalism. Typical AI−generated styling (blue theme, sparse, flat) OR webpage template vibes. Lacks formality,
    authority, or credibility.
1: Unprofessional. Obviously generic AI output or inappropriate format. No professional character or document authenticity.
</rubric>

<output_format>
Respond in JSON:
{
  "reason": "Describe the document's styling and explain whether it is generated by professional human editor and conveys
      formality, authority, and professional authenticity.",
  "score": <integer 1−5>
}
</output_format>

## C. Additional Analysis

### C.1. Component Ablation

The main results in Section 6 ablate the refinement loop; here we isolate the remaining components on a randomly sampled subset of 50 examples from RICHDOCBENCH (Table 6). Syntax rollback shows the largest impact on visual quality: removing it causes rendering correctness, layout appropriateness, and professionalism to all decline, confirming that automatic rollback of invalid edits prevents cascading formatting errors that would otherwise accumulate during multi-step editing.

The task planner contributes more modestly, with small but consistent drops across all dimensions when removed, suggesting that explicit subgoal tracking helps coordinate long-horizon editing without being individually decisive. Template retrieval presents a more nuanced trade-off: removing it reduces content alignment (4.38→4.24), as the model

*Table 6.* Component ablation on RICHDOCBENCH. Each row removes one component from the full system.

| Variant | Content Align. | Rendering | Layout | Prof. |
|---|---|---|---|---|
| FORMACT | 4.38 | 4.58 | 3.62 | 3.84 |
| w/o Task Planner | 4.32 | 4.54 | 3.62 | 3.82 |
| w/o Syntax Rollback | 4.40 | 4.50 | 3.55 | 3.78 |
| w/o Template Retrieval | 4.24 | 4.68 | 3.70 | 3.82 |

loses access to structural references from real-world documents, but slightly improves rendering and layout because the resulting outputs are structurally simpler and less prone to formatting issues inherited from complex templates. This observation is consistent with the professionalism progression from *DG* (3.75) to *TAG* (3.80) to FORMACT (3.92) in Table 1, where templates contribute meaningfully but their benefit depends on how they are used. The template pool sensitivity analysis in Section C.3 further explores this trade-off. Finally, edit-triggered context compression cannot be directly ablated, as 82% of samples would exceed the context window limit without it; its contribution is instead characterized in the cost analysis below.

## C.2. Cost and Runtime

Table 7 reports the per-phase cost and runtime of FORMACT, averaged over 50 samples. The initial generation phase completes in approximately 3.6 minutes at $0.47, using about 281K prompt tokens with a 54.8% cache hit rate. During refinement, prompt tokens grow to approximately 766K by round five, but the cache hit rate rises above 72%

*Table 7.* Per-phase cost and runtime of FORMACT.

| Phase | Time | Cost | LLM Calls | Cache% |
|---|---|---|---|---|
| Generation (R1) | 3.6 min | $0.47 | 12.4 | 54.8% |
| Refinement (R2–R6, avg) | 4.0 min | $0.56 | 15.5 | 72.7% |
| Total (per document) | 23.6 min | $3.28 | 90.1 | – |

as repeated context patterns are reused, keeping per-round cost stable at an average of $0.56. Refinement is capped at five rounds, but the reviewing agent can terminate early when no issues are found, so not all documents require the full five rounds. The total cost averages $3.28 per document. Edit-triggered context compression is critical to enabling this: it achieves an 84% character-level reduction, keeping the working context bounded so that per-round cost does not grow proportionally with the number of iterations. Without compression, 82% of samples would exceed the context window limit, making multi-round refinement infeasible. In professional document authoring, where a single rendering defect can make a document unsuitable for distribution, this cost represents a practical trade-off that automates what would otherwise require significant manual iteration.

## C.3. Template Retrieval Sensitivity

The component ablation above reveals a trade-off between template richness and rendering complexity: templates provide realistic document structures that improve content alignment, but their structural complexity can introduce formatting issues that are harder to render correctly. Here we further examine this trade-off by varying the template pool size (Ta-

*Table 8.* Impact of template pool configuration on document quality.

| Variant | Content Align. | Rendering | Layout | Prof. |
|---|---|---|---|---|
| FORMACT (full pool) | 4.38 | 4.58 | 3.62 | 3.84 |
| 10% template pool | 4.38 | 4.48 | 3.84 | 3.92 |
| w/o template retrieval | 4.24 | 4.68 | 3.70 | 3.82 |

ble 8). Without retrieval, content alignment drops (4.38→4.24) as the model loses access to real-world document structures, but rendering correctness reaches its highest value (4.68) because the model produces structurally simpler outputs with fewer opportunities for formatting errors. This pattern is consistent with *Codex* + refinement in Table 2, which also achieves strong rendering on simpler outputs. Interestingly, a 10% pool improves professionalism (3.84→3.92) and layout (3.62→3.84) over the full pool, suggesting that a smaller pool reduces retrieval noise and surfaces more relevant templates, leading to cleaner adaptation. These results indicate that template pool quality matters more than quantity, and that principled pool curation can help balance this richness–complexity trade-off.

## C.4. Qualitative Comparison

Figure 4 shows the first rendered page from each method for the query: *"Create a 'Safety Certificate Application Form 2013' for sports grounds under the Safety of Sports Grounds Act 1975 and the Fire Safety and Safety of Places of Sport Act 1987."* This example illustrates the key differences across methods. *DG* produces a generic layout with default blue

styling that does not reflect the formal character of a government application form. *Codex* generates a similar document that still lacks the dense field layout and form-specific styling expected of this document type. *TAG* retrieves a template that does not match the query well, resulting in a layout that, while visually structured, is not well adapted to the specific form requirements. *Docx Agent* produces the simplest output with minimal styling, reflecting the expressiveness constraints of the `python-docx` library. In contrast, FORMACT produces the most professional result: it features styled header elements, properly spaced form fields with fill lines, multi-column table layouts, and pagination, closely resembling a real-world government application form.

(a) *DG*      (b) *Codex*      (c) *TAG*

(d) *Docx Agent*      (e) FORMACT

*Figure 4.* Qualitative comparison of rendered first pages across methods for a safety certificate application form query. FORMACT produces the most professional output with structured form fields, styled headers, and proper pagination.

## C.5. RichDocFuzz Example

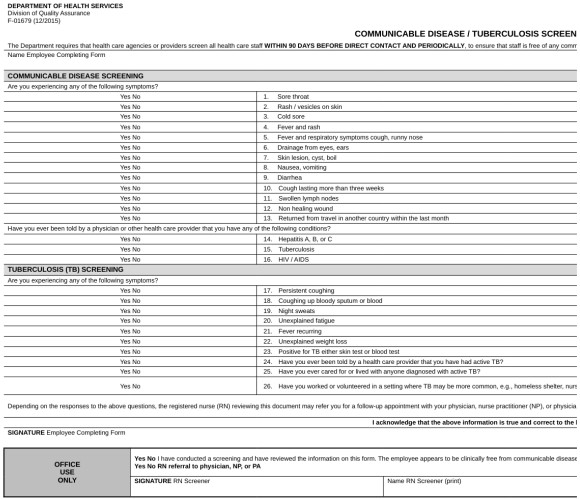

*Figure 5.* An example from RICHDOCFUZZ with a B2 (table overflow) perturbation: text extends beyond the right edge of the table, getting cut off at the cell boundary. Such defects are only visible on the rendered page.

