# OpenReview forum: "FormAct: Agentic Source Editing for Rich-Format Document Generation"
_ICML.cc/2026/Conference — ICML 2026 regular_

### Official Review · Reviewer_SbHh · 2026-03-01

**Soundness:** 3
**Presentation:** 2
**Significance:** 3
**Originality:** 2
**Overall Recommendation:** 4
**Confidence:** 4

**Summary:**

The paper frames rich-format document generation as a document engineering problem and proposes FORMACT, an agentic HTML source-editing system with iterative render–review refinement. The article seeks to study the central issue of how to reliably produce professional, printable documents from short natural language queries rather than just generic HTML or webpage-like outputs.

The system combines template retrieval from a large Word-to-HTML corpus, a syntax-aware line-based HTML editor with rollback, a task planner, and a multimodal reviewer operating on rendered pages. The refinement loop is positioned as the key driver for rendering correctness and layout quality.

In addition, the paper introduces two benchmarks: RichDocBench for end-to-end generation and RichDocFuzz for formatting error detection by VLM reviewers. Overall, the authors analyze an important concept: whether source-level iterative editing with visual feedback leads to more reliable document formatting than one-shot or API-based approaches.

**Compliance With Llm Reviewing Policy:**

Affirmed.

**Key Questions For Authors:**

1. How much of the improvement comes from the FORMACT architecture versus the strength of the proprietary LLM used? Have you evaluated the same framework with smaller or open-source models?

2. Since both templates and evaluation data come from Word-derived HTML, how well does the system generalize beyond this distribution? Have you tested on stylistically different or out-of-domain documents?

3. What is the average number of LLM calls, token usage, and latency per document compared to single-pass baselines? Is the improvement in rendering correctness justified by the additional compute cost?

**Limitations:**

The authors did not include an limitation section in the paper. The paper can benefit from including more limtation discussion on the framwork design and cost efficiency analysis.

**Strengths And Weaknesses:**

## Strengths

1. The problem is well-motivated and clearly scoped: professional, print-oriented rich documents are genuinely underexplored compared to web layout or slide generation. Treating document generation as source-level engineering rather than pure text generation is conceptually clean and practically meaningful.

2. The system design is coherent and internally consistent. The syntax-aware patching, rollback mechanism, and edit-triggered context compression are thoughtful engineering decisions that directly address stability issues in multi-round editing.

3. The evaluation is relatively comprehensive within its scope. The separation of content alignment, rendering correctness, layout appropriateness, and professionalism, plus human preference studies, gives a reasonably convincing picture that iterative refinement improves presentation quality.

## Weaknesses

1. The work is heavily engineering-driven and does not introduce fundamentally new modeling ideas. The backbone is still a strong proprietary LLM, and it is unclear how much of the gain comes from architecture versus raw model capability.

2. Evaluation is confined to their own constructed benchmark, which raises generalization concerns. There is no evidence that the system transfers beyond Word-derived HTML distributions or to more diverse document styles.

3. The cost-performance tradeoff is not analyzed. Multi-round editing, rendering, and multimodal review are likely expensive, and the paper does not quantify token usage, latency, or scalability relative to simpler baselines.

---

> ### Author Rebuttal · Authors · 2026-03-31
>
> Dear Reviewer SbHh,
>
> We thank the reviewer for the thoughtful review, and for recognizing the practical significance of source-level document engineering and the coherent system design. We address each concern below.
>
> **Q1: Architecture vs. Model Capability**
>
> All methods in our experiments, including all baselines, use the same underlying model (GPT-5.2). The gap between Codex and FormAct (e.g., 4.39 vs. 4.81 on rendering correctness, 3.72 vs. 3.92 on professionalism) demonstrates that the gains come from FormAct's specialized tools and workflow rather than model capability alone. A strong model applied naively to document generation does not automatically produce professional formatting, which is one of our key empirical findings. We also note that FormAct requires two crucial capabilities from the underlying model: source editing ability for the editing agent and visual understanding for the review agent. Our earlier experiments with Qwen3-8B showed that these capabilities were insufficient to support the pipeline effectively. We agree that further investigation with larger open-source models would be valuable to better understand the minimum capability threshold required.
>
> **Q2: Generalization Beyond Word-Derived HTML**
>
> While the documents originate from Word-to-HTML conversion, the underlying corpus of roughly three million documents covers a wide range of real-world document types including reports, legal and compliance documents, proposals, and forms. We collected as broadly as possible from publicly available sources to maximize diversity. We chose Microsoft Word documents as the source because Word remains the most widely used document authoring tool in professional settings, making it a representative proxy for the types of rich-format documents users produce in practice. We acknowledge that extending the evaluation to other sources would further strengthen the generalizability evidence, and will discuss this in the limitations section of the revised manuscript.
>
> **Q3: Cost and Runtime Analysis**
>
> We appreciate this practical question. FormAct completes the initial generation in about 3.6 minutes with 12.4 LLM calls (0.47 USD), using approximately 281K prompt tokens (54.8% cached). Each subsequent refinement round takes roughly 3.4–4.6 minutes with 14–16 calls (0.42–0.63 USD), with prompt tokens growing to 766K by round 5 but maintaining above 72% cache hit rate. We cap refinement at a maximum of 5 rounds, totaling approximately 3.28 USD per document.  Edit-triggered context compression is critical for enabling this: without it, 82% of samples would exceed the context window limit, but the 84% compression rate keeps the working context bounded. We will include the full per-round statistics in the revised manuscript.
>
> In professional settings, a single rendering defect can make a document unsuitable for distribution, and manually iterating on formatting is itself a time-consuming process. We believe the additional compute is a practical investment that automates what would otherwise require significant human effort.
>
> We are grateful for the reviewer's constructive feedback, which has helped us identify important areas to strengthen. We hope our responses address the concerns raised and welcome any further questions.
>
> Sincerely,
>
> The Authors

---

> > ### Author Rebuttal · Reviewer_SbHh · 2026-04-05
> >
> > 1. Although the authors claim that they tried to cover millions of documents in their own proposed benchmark, the generalization concern still exists as it's about the data distribution instead of the quantity.
> > 2. For the cost analysis, the authors only report costs of their method, without comparing the cost of the baselines like codex, etc.
> >
> > Besides these, my concern in Q1 has been resolved. Therefore, I would say my concern is partially resolved and will maintain my score

---

> > > ### Author Response · Authors · 2026-04-06
> > >
> > > Dear Reviewer SbHh,
> > >
> > > Thank you for the follow-up and for confirming the resolution of Q1. We apologize for not addressing the remaining concerns clearly enough in our initial response.
> > >
> > > **Q2: Concerns about Generalization**
> > >
> > > We understand the concern is about generalization rather than document quantity. We emphasize the scale of documents to highlight our effort in building a representative corpus of professional document authoring. We chose Microsoft Word as the source because it remains the most widely used authoring tool in professional settings, serving as a reliable proxy for the documents users produce in practice. The template pool is designed to provide broad coverage across common document types.
> > >
> > > For queries that fall outside the template distribution, FormAct's retrieval identifies the most similar template and adapts it through iterative source editing, which is precisely what the editing and review agents are designed for. We acknowledge that performance on long-tail cases may not match well-covered document types, and our current work focuses on common, high-frequency scenarios which account for the majority of real-world document authoring needs as a first step. We will discuss this as a limitation in the revised manuscript.
> > >
> > > **Q3: Cost/Runtime Comparisons With Baselines**
> > >
> > > | Method | Latency | Calls | Cost |
> > > |---|:---:|:---:|:---:|
> > > | DG | 1.9 min | 1 | 0.25 USD |
> > > | TAG | 1.8 min | 1 | 0.34 USD |
> > > | Docx Agent | 1.1 min | 4.3 | 0.10 USD |
> > > | Codex | 3.2 min | 3.4 | 0.60 USD |
> > > | FormAct (Gen.) | 3.6 min | 12.4 | 0.47 USD |
> > > | FormAct (Refine avg per round) | 4.0 min | 15.5 | 0.56 USD |
> > > | FormAct (Total) | 23.6 min | 90.1 | 3.28 USD |
> > >
> > > The generation cost is comparable across methods. DG is the single-pass baseline; TAG adds slightly more cost from template input tokens. Docx Agent is cheapest due to the limited expressiveness of library-based APIs. FormAct's generation is slightly more expensive than TAG due to multi-turn tool use, but remains cheaper than Codex. As shown in Table 1(a), FormAct without refinement already achieves strong results, but rendering correctness and reliability can be further improved through refinement rounds, where each round costs approximately the same as generation.
> > >
> > > We believe this cost is justified. In the software engineering field, practitioners have shifted from single-pass code generation to long-horizon autonomous agents (e.g., Codex, Claude Code), accepting higher compute for substantially improved reliability. We argue the same applies here: professional document authoring involves significant human effort into formatting iterations, and automating this is a practical trade-off. Ensuring that the additional compute translates into real productivity gains and high-quality deliverables is an important direction, and we believe FormAct demonstrates a meaningful step in this regard. This is precisely why we frame our work as document engineering.
> > >
> > > Once again, we thank the reviewer for your constructive feedback. We hope our clarifications and additions can address your concerns.
> > >
> > > Sincerely,
> > >
> > > The Authors

---

### Official Review · Reviewer_PF3A · 2026-03-11

**Soundness:** 3
**Presentation:** 4
**Significance:** 4
**Originality:** 2
**Overall Recommendation:** 4
**Confidence:** 4

**Summary:**

This paper studies automated generation of rich-format documents from natural language instructions by directly editing the underlying HTML source. The authors propose FORMACT, an agentic framework that iteratively edits document source, renders the document, and refines it based on feedback from a reviewer agent. The system combines template retrieval from a large corpus of converted Word documents, a syntax-aware HTML editing environment, and a render–review refinement loop. The paper also introduces RICHDOCBENCH, a benchmark of rich-format document generation tasks, and RICHDOCFUZZ, a dataset for evaluating document rendering error detection. Experimental results show improvements in rendering correctness, layout quality, and human preference compared to several generation and agent-based baselines.

**Compliance With Llm Reviewing Policy:**

Affirmed.

**Final Justification:**

I mark my concerns to be resolved and maintain my weak accept score, owing to the core limitations I also pointed out due to which I am not supporting a full accept score.

**Key Questions For Authors:**

1) Much of the automatic evaluation relies on LLM-based judging (e.g., GPT-5.2 rubric scoring). While the paper introduces RICHDOCFUZZ to study reviewer capabilities, the reported reviewer accuracy (~70% overall) suggests that the reviewer itself may miss certain errors, particularly subtle styling defects. Could the authors elaborate on how sensitive the final evaluation results are to reviewer reliability? For example, do the conclusions remain consistent if different judge models or alternative evaluation strategies are used?
2) Rendering correctness is one of the primary claimed advantages of FORMACT. However, the current evaluation mainly uses rubric-based scoring. Have the authors considered incorporating more objective rendering metrics, such as automatic detection of overflow, clipping, malformed HTML structure, or DOM inconsistencies? Such measures might provide stronger evidence that the method improves structural correctness.
3) The system relies on an iterative editing–review–refinement loop. Could the authors provide additional analysis of the average number of refinement iterations, total inference cost, and latency relative to single-pass or simpler generation approaches? Understanding the practical computational trade-offs would help assess the feasibility of deploying the system in real-world document authoring workflows.
4) The system uses template retrieval from a large corpus of documents to guide generation. How sensitive is the overall performance to retrieval quality? For instance, what happens if retrieval returns less relevant templates, or if the template pool is reduced in size? An ablation examining the impact of retrieval quality could clarify how much of the performance gain comes from the agentic editing process versus strong retrieval initialization.
5) The current formulation operates on HTML source derived from Word documents. How well would the approach generalize to other document representations (e.g., LaTeX, PDF layout structures, or other structured document formats)? Are there aspects of the system - such as the editing environment or reviewer model that rely heavily on HTML-specific properties?
6) The baselines primarily include prompting approaches and general agent frameworks. Could the authors comment on how FORMACT might compare to more specialized layout or document-generation systems (e.g., systems designed specifically for structured layout synthesis)? Clarifying this positioning would help contextualize the results relative to existing document generation methods.
7) Figure 2 suggests that refinement generally improves rendering correctness but that layout appropriateness may peak at intermediate iterations. Could the authors elaborate on why later refinement stages sometimes degrade layout quality? Understanding the failure modes of the refinement loop would help clarify the limitations of the approach.

**Limitations:**

Yes

**Strengths And Weaknesses:**

The paper addresses a practical and underexplored problem: generating complex structured documents where content and layout must both be correct. Many failures in this setting only appear after rendering, making the iterative render–review–edit paradigm well motivated. The system architecture is clearly presented, and the workflow diagram communicates the interaction between editing agent, rendering environment, and reviewer agent effectively.
A notable design decision is to operate directly on HTML source editing rather than application interfaces or image outputs. This representation allows incremental patches, syntax checking, and rollback of invalid edits. The proposed edit-triggered context compression is also a thoughtful mechanism to prevent stale grounding after edits change line numbering. The dataset and benchmark contributions are meaningful. RICHDOCBENCH contains over 34k training examples, providing a standardized evaluation task for structured document generation.
Experimental results are generally consistent with the claimed advantages of the system. FORMACT achieves the best scores on rendering correctness and professionalism, which are precisely the dimensions the refinement loop aims to improve. Human preference evaluations also show a clear advantage over competing approaches.
Finally, the reviewer reliability experiments using RICHDOCFUZZ are a useful addition. They demonstrate that vision-language models can detect many document issues, while also revealing limitations in detecting subtle styling problems.

I will be listing out some concerns below:
The methodological novelty is moderate. The overall architecture - generation followed by critique and refinement, is similar to many recent agentic frameworks. The main contribution lies in adapting this paradigm to document source editing rather than introducing fundamentally new modeling ideas. The evaluation relies heavily on LLM-based judging, particularly for rubric scoring. While human preference studies are included, they are relatively small in scale. More objective metrics or larger human studies would strengthen the empirical claims.
Baseline comparisons could also be stronger. While the paper includes several prompting and agent baselines, it is less clear how FORMACT compares against more specialized document-generation or layout-generation systems. Finally, the benchmark relies on Word-to-HTML conversions, which may introduce biases or artifacts specific to the conversion pipeline. While the authors acknowledge this issue, it somewhat limits the generality of the benchmark.

I like the paper but I also have some important concerns that I have listed in the questions section and would like to see more discussions around those, which may impact my scores.

---

> ### Author Rebuttal · Authors · 2026-03-31
>
> Dear Reviewer PF3A,
>
> We thank the reviewer for the thorough review and for recognizing the practical significance of the problem and the effectiveness of our system design. We address each concern below.
>
> **Q1: Evaluation Robustness**
>
> We re-ran rubric grading on 50 sampled instances with three additional independent judges:
>
> | Method | Opus 4.5 | GPT-5.2 | GPT-5.4 | Gemini 3 Flash |
> |---|:---:|:---:|:---:|:---:|
> | FormAct | **3.83** | **3.97** | **4.09** | **4.92** |
> | TAG | 3.59 | 3.67 | 3.67 | 4.57 |
> | DG | 3.56 | 3.57 | 3.81 | 4.71 |
> | Codex | 3.51 | 3.51 | 3.69 | 4.61 |
> | Docx Agent | 3.12 | 3.35 | 3.42 | 4.21 |
>
> *Table: Visual Quality Overall (average of rendering correctness, layout appropriateness, and professionalism).*
>
> FormAct ranks first with all four judges on visual quality overall, and the same holds on individual dimensions, which supports the reliability of our evaluation. We also experimented with pairwise comparison but observed significant positional bias in multi-page document comparisons, leading us to adopt per-dimension rubric scoring instead. Regarding reviewer agent sensitivity, replacing the GPT-5.2 reviewer with GPT-4.1 (38.0% vs. 70.0% on RichDocFuzz) resulted in rendering correctness dropping from 4.58 to 4.14. As VLM visual understanding continues to improve, we expect the refinement loop to become increasingly effective.
>
> **Q2: Objective Rendering Metrics**
>
> FormAct's syntax-aware editor already validates HTML structure and rolls back invalid edits. However, many formatting issues only manifest in the rendered output and are not detectable from source code alone, which is why we assess rendering quality through VLM-based visual review.
>
> **Q3: Cost and Runtime**
>
> FormAct completes initial generation in ~3.6 min (0.47 USD), each refinement round in ~3.4–4.6 min (0.42–0.63 USD), capped at 5 rounds, totaling ~3.28 USD/document. Context compression is critical: without it, 82% of samples would exceed the context window. The 84% compression rate keeps per-round cost stable. For professional document authoring, where a single rendering defect can make a document unsuitable for distribution, we believe this is a worthwhile trade-off.
>
> **Q4: Impact of Template Retrieval Quality**
>
> In general, a larger knowledge base can improve coverage for new samples, but it may also increase the difficulty of accurate retrieval. Following your suggestion, we evaluated FormAct with a randomly sampled 10% subset and with no retrieval:
>
> | Variant | Content Align. | Professionalism |
> |---|:---:|:---:|
> | FormAct (full pool) | 4.38 | 3.84 |
> | FormAct (10% pool) | 4.38 | 3.92 |
> | w/o Template Retrieval | 4.24 | 3.82 |
>
> *Table: Impact of template retrieval settings on document quality.*
>
> As shown, when no retrieval is used, content alignment drops significantly, and professionalism also decreases to some extent. With only a 10% random sample of the template pool, retrieval accuracy likely improves due to reduced noise, leading to a clear improvement in professionalism. We acknowledge that template pool curation is an important problem that could yield further improvements, and will discuss this in the revision.
>
> **Q5: Generalization to Other Formats**
>
> FormAct's core design is not specific to HTML. The review agent operates on rendered pages and is format-agnostic. The editing agent can also adapt to other markup languages such as LaTeX. We chose HTML over OOXML because LLMs are extensively pre-trained on HTML, whereas OOXML is verbose with many interdependent components. Though extending FormAct to other formats is a natural direction, constructing high-quality benchmarks for each requires substantial effort, and we will discuss this in the limitations section.
>
> **Q6: Comparison to Specialized Systems**
>
> Prior works adopt fundamentally different setups. DocEdit [1] and DocEdit-v2 [2] perform localized edits on existing documents. Layout methods like PosterLLaMA [3] focus on spatial arrangement of predefined elements on a single page. Our setting requires generating complete, multi-page documents from a short query. No existing specialized system supports this end-to-end setting, which is why we adopt baselines reflecting realistic alternatives.
>
> **Q7: Layout Degradation at Later Stages**
>
> Rendering correctness trends upward for all methods except TAG, as the reviewer agent reliably catches concrete defects. Layout appropriateness tends to fluctuate: fixing rendering defects may shift spacing or alignment. This is expected given that layout quality is sensitive to interacting spatial features. While FormAct achieves the best layout scores overall, this suggests room for improvement in fine-grained spatial reasoning during refinement.
>
> We thank the reviewer for the valuable questions. We are happy to discuss any remaining concerns.
>
> Sincerely,
>
> The Authors
>
> **References**
> 1. Mathur et al. DocEdit. AAAI 2023.
> 2. Suri et al. DocEdit-v2. EMNLP 2024.
> 3. Seol et al. PosterLLaMA. ECCV 2024.

---

> > ### Author Rebuttal · Reviewer_PF3A · 2026-04-04
> >
> > Thank you for the detailed rebuttal and additional analysis. The clarifications on evaluation robustness, retrieval impact, and system cost help address my key concerns.
> >
> > There are still some limitations, particularly around evaluation dependence and overall novelty, but these are now clearly acknowledged and better contextualized.
> >
> > Overall, I consider my concerns to be sufficiently addressed and am happy to keep my original scores.

---

> > > ### Author Response · Authors · 2026-04-06
> > >
> > > Dear Reviewer PF3A,
> > >
> > > Thank you for the constructive engagement. We are glad that our rebuttal was able to address your concerns.
> > >
> > > We would like to briefly reinforce that the core contribution of this work lies in identifying and formalizing end-to-end rich-format document generation as a document engineering problem that inherently requires iterative refinement driven by test signals. Much like software engineering, where bugs only surface when code is executed against test cases, formatting defects in rich-format documents only become apparent when the source is rendered. Our system operationalizes this insight through source-level editing that localizes fixes to specific defects without disturbing surrounding content, enabling a practical and composable refinement loop.
> > >
> > > Once again, we sincerely appreciate your expertise and time, and will incorporate the additional experiments along with all the analyses prompted by your concerns into the revision.
> > >
> > > Sincerely,
> > >
> > > The Authors

---

### Official Review · Reviewer_v61G · 2026-03-12

**Soundness:** 3
**Presentation:** 3
**Significance:** 3
**Originality:** 2
**Overall Recommendation:** 4
**Confidence:** 2

**Summary:**

This paper studies end-to-end generation of rich-format documents from short natural-language requests, rather than editing an existing document. The proposed system, FORMACT, treats the problem as source-level document engineering: an editing agent operates directly on HTML with syntax-aware view/edit tools, a template retriever initializes the process from a pool of professionally authored documents, and a review agent critiques rendered pages to guide iterative refinement.

**Compliance With Llm Reviewing Policy:**

Affirmed.

**Key Questions For Authors:**

The automatic rubric grading uses GPT-5.2, and the paper also states that GPT-5.2 is the underlying generator for all methods. Can the authors provide a stronger judge-robustness analysis, for example with multiple independent judges or human scoring on each rubric dimension? A strong positive answer would increase my confidence in the main empirical claims.

The paper ablates refinement, but not several other core components such as template retrieval, edit-triggered context compression, syntax-aware rollback, or the task planner. Can the authors provide ablations isolating these components, along with cost/runtime statistics? If some of these components turn out to be essential, that would strengthen the paper considerably.

Since RichDocBench queries are derived by back-translation from documents in the same overall corpus, what safeguards were used against near-duplicate leakage or retrieval-style matching between the template pool and the benchmark? Please quantify overlap and provide copy-rate or novelty analyses. A convincing answer here would materially improve my view of the evaluation.

The reviewer-agent validation tops out at 70% overall accuracy and appears notably weaker on styling failures. How sensitive is FORMACT to reviewer quality in the full pipeline, and what failure modes are most common when the reviewer is wrong? A clearer answer would help me judge whether the review-guided loop is robust enough in practice.

The dataset appears to be built from roughly three million publicly available, human-authored Word documents. Please clarify data provenance, copyright/licensing status, PII filtering, and whether any dataset/templates will be released. This would affect both my reproducibility assessment and my ethics assessment.

**Limitations:**

No. The paper includes a brief impact statement, but the limitations discussion is not yet adequate. In particular, the authors should discuss at least: (i) possible benchmark bias introduced by deriving queries from the same document collection used to build the template pool; (ii) the limited reliability of VLM-based review, especially for styling defects; (iii) dependence on a strong proprietary model stack; and (iv) data provenance, copyright, and privacy issues arising from the use of millions of public user-authored documents.

**Strengths And Weaknesses:**

Strength:

The paper has several clear strengths. First, the problem setting is meaningful and underexplored: most prior work assumes an existing rich document to edit, whereas this work targets generation from scratch and does so in a source-editing setting rather than through a narrow application API. That framing is well motivated, and the analogy to software engineering agents is sensible. Second, the system design is coherent. The combination of direct HTML editing, syntax-checked rollback, retrieval over template specifications, and render-review-refine feedback is intuitively appropriate for a task where many failures only become visible after rendering. Third, the benchmark effort is substantial: the paper constructs RichDocBench from a large pool of public Word documents, uses a 34,188/500 train-test split, and adds RichDocFuzz to probe reviewer reliability. Fourth, the empirical results are promising: FORMACT achieves the strongest rendering correctness and layout appropriateness among reported baselines, and human evaluators rank it first much more often than competing methods.

Weakness:

The evaluation is not yet as convincing as I would like for the paper’s strongest claims. Automatic grading uses GPT-5.2 as the judge, and the paper also states that GPT-5.2 is the underlying generator for all methods and evaluation; this judge-generator overlap raises concern about bias or at least limited robustness of the reported rubric scores. The human study is helpful, but it is run on only 50 sampled instances, in two separate 3-way comparisons, so it is still relatively small given the scope of the claims. In addition, the reviewer-agent validation on RichDocFuzz reaches only up to 70% overall accuracy, with substantially worse performance on styling-related failures, which makes me less convinced that the visual reviewer is already reliable enough to support a strong general conclusion about review-guided refinement.

The paper shows an ablation removing refinement, which is useful, but the system has several other central components—template retrieval, task planning, syntax-aware editing/rollback, and especially edit-triggered context compression—and these are not isolated convincingly in the main empirical analysis. Because of this, it is hard to know which ingredients are essential versus merely helpful. Similarly, the comparison budget is not obviously matched: FORMACT can adaptively inspect many templates and perform iterative tool use, while TAG is a much weaker in-context baseline and Docx Agent is explicitly disadvantaged by the fact that the shared prompt constraints are HTML-specific and do not apply to it. This does not invalidate the comparison, but it weakens the strength of the causal claims about where the gains come from.

The paper is mostly clear and the high-level narrative is easy to follow. Figure 1 is helpful, and the separation between source editing and rendered-page review is well explained. The benchmark construction and evaluation protocol are also described more carefully than in many systems papers. However, the paper still has some polish issues: there are a number of grammatical rough edges, at least one incomplete appendix reference (“see Appendix .”), and some implementation details that are too thin for full reproducibility, especially around the reward model, template filtering thresholds, exact prompting/tool orchestration, stopping criteria, and the precise instantiation of the review agent. These are fixable, but they matter for a paper whose main contribution is a system pipeline.

I think the topic is important and practical. Rich-format document generation is a real workflow problem, and moving from “edit an existing document” to “generate a professional document from scratch” is a useful shift. The benchmark contribution may also be valuable for future work. My hesitation is that the impact currently feels somewhat specialized, and the evidence does not yet fully establish that the method generalizes beyond the benchmark construction pipeline. In particular, the benchmark queries are derived by back-translation from the same corpus of documents that also supplies the training split and template pool, so I would like a much more careful discussion of distribution overlap, near-duplicate control, and whether the gains partly reflect retrieval-friendly benchmark construction. That concern lowers my assessment of significance somewhat, not because the problem is unimportant, but because the experimental setup may be unusually favorable to the proposed approach.

The paper is moderately original but not highly so. The main novelty comes from adapting source-editing agents to rich-format document generation, integrating rendering-based review with syntax-aware HTML patching, and introducing the two datasets. These are meaningful contributions. At the same time, the overall method is largely a careful composition of already-familiar ingredients—tool-using agents, retrieval, iterative refinement, and multimodal review—rather than a fundamentally new algorithmic idea. I therefore view originality as fair-to-good rather than excellent.

---

> ### Author Rebuttal · Authors · 2026-03-31
>
> Dear Reviewer v61G,
>
> We are grateful for the detailed review and for acknowledging the coherent system design and benchmark effort. We address each concern below.
>
> **Q1: Judge-Generator Overlap and Evaluation Robustness**
>
> We re-ran rubric grading on 50 sampled instances with three additional independent judges:
>
> | Method | Opus 4.5 | GPT-5.2 | GPT-5.4 | Gemini 3 Flash |
> |---|:---:|:---:|:---:|:---:|
> | FormAct | **3.83** | **3.97** | **4.09** | **4.92** |
> | TAG | 3.59 | 3.67 | 3.67 | 4.57 |
> | DG | 3.56 | 3.57 | 3.81 | 4.71 |
> | Codex | 3.51 | 3.51 | 3.69 | 4.61 |
> | Docx Agent | 3.12 | 3.35 | 3.42 | 4.21 |
>
> *Table: Visual Quality Overall (average of rendering correctness, layout appropriateness, and professionalism).*
>
> FormAct ranks first with all four judges. The same holds on individual dimensions: first on layout and professionalism with all four, and first on rendering with three out of four. The strong inter-annotator agreement and alignment with all four VLM judges suggest reliable results. Full per-dimension results will be included in the revision.
>
> **Q2: Component Ablations and Runtime/Cost**
>
> The w/o Refinement ablation in Table 1(a) already shows that removing refinement significantly reduces rendering correctness. The progression DG (3.75) → TAG (3.80) → FormAct (3.92) on professionalism, and DG (2.48) → TAG (2.02) → FormAct (1.50) on human preference rank, shows templates contribute meaningfully. We have conducted additional experiments to isolate the remaining components:
>
> | Variant | Content Align. | Rendering | Layout | Prof. |
> |---|:---:|:---:|:---:|:---:|
> | FormAct | 4.38 | 4.58 | 3.62 | 3.84 |
> | w/o Task Planner | 4.32 | 4.54 | 3.62 | 3.82 |
> | w/o Syntax Rollback | 4.40 | 4.50 | 3.55 | 3.78 |
> | w/o Template Retrieval | 4.24 | 4.68 | 3.70 | 3.82 |
>
> *Table: Component ablation*
>
> Syntax rollback shows the largest visual quality drop (rendering: 4.58→4.50, layout: 3.62→3.55, prof.: 3.84→3.78), confirming it prevents cascading errors. Task planner removal causes modest but consistent drops. Removing templates reduces content alignment (4.38→4.24) as the model loses access to real-world document structures as references, but slightly improves rendering and layout due to structurally simpler outputs, consistent with Codex + refinement in Table 2. For edit-triggered context compression, a direct ablation is not feasible as 82% of samples would exceed the context window limit without it. We will include the full ablation analysis in the revised manuscript.
>
> Regarding cost and runtime, FormAct completes initial generation in ~3.6 min (0.47 USD), each refinement round in ~3.4–4.6 min (0.42–0.63 USD), capped at 5 rounds, totaling ~3.28 USD/document. Context compression achieves 84% reduction, keeping per-round cost stable.
>
> **Q3: Data Leakage Between Template Pool and Test Set**
>
> We acknowledge that calling the template pool a "training dataset" may have caused confusion. We selected 34,188 examples covering common templates across various domains. Once extracted, the pool functions as an independent knowledge base. Selecting from it is analogous to how a human author chooses from a standard template library, and should not be considered data leakage. To ensure our evaluation is not influenced by this design, we assess all rubric dimensions independently without using human-authored documents as gold references.
>
> A key challenge in our work is selecting an appropriate template for each query, then adapting new content while fixing rendering errors and layout issues through fine-grained edits. This is precisely where FormAct's editing agent and review agent contribute, and these stages are not simply reproducing existing templates. We acknowledge that our paper could have made this distinction clearer, and will clarify in the revised manuscript.
>
> **Q4: Reviewer Agent Reliability**
>
> VLM-based visual review remains an open challenge. The Remote Labor Index [1] identifies visual understanding as a key limitation of current AI agents. Our RichDocFuzz results show clear improvement with model capability: GPT-4.1 achieves 38.0% accuracy while GPT-5.2 reaches 70.0%, suggesting that reviewer reliability will continue to improve with future models. We replaced the GPT-5.2 reviewer with GPT-4.1, resulting in rendering correctness dropping from 4.58 to 4.14. In practice, the failure modes observed are mostly insensitivity to subtle spacing adjustments rather than outright errors.
>
> **Q5: Data and Code Release**
>
> All documents are from publicly available sources. We are undergoing internal compliance review and plan to release RichDocBench, RichDocFuzz, and FormAct code upon completion.
>
> We hope our clarifications address the concerns raised and welcome further discussion.
>
> Sincerely,
>
> The Authors
>
> **References**
>
> 1. Mazeika et al. Remote Labor Index: Measuring AI Automation of Remote Work. arXiv:2510.26787, 2025.

---

### Official Review · Reviewer_9REF · 2026-03-12

**Soundness:** 3
**Presentation:** 3
**Significance:** 3
**Originality:** 3
**Overall Recommendation:** 4
**Confidence:** 3

**Summary:**

This paper proposes FORMACT, an agentic system based on HTML source editing. It formulates rich-format document generation as a document engineering task and generates rich-format documents from user queries through multi-round iterative optimization. Additionally, the authors construct two benchmarks, RICHDOCBENCH and RICHDOCFUZZ, to systematically evaluate the quality of generated documents and the formatting-error recognition capabilities of VLMs, respectively. Experimental results and human-preference studies demonstrate that the proposed FORMACT achieves superior performance compared to both single-pass and multi-pass baselines.

**Compliance With Llm Reviewing Policy:**

Affirmed.

**Final Justification:**

This paper puts forward an innovative document engineering task, and constructs two valuable benchmarks for the community. To this end, we lean toward the positive side.

**Key Questions For Authors:**

Please refer to Weaknesses 1-3.

**Limitations:**

yes

**Strengths And Weaknesses:**

### Strengths
1. The perspective of transforming rich-format document generation into a document engineering task, which utilizes dual feedback from code editing and visual rendering, is inspiring and innovative.

2. Compared to existing automated document editing and generation methods, the proposed FORMACT does not require an initial draft and edits the HTML source code directly, enabling much broader applicability and greater flexibility.

3. The construction of the two benchmarks (RichDocBench and RichDocFuzz) provides valuable resources and lays a foundation for automated evaluation and future research in this domain.

4. The experimental results convincingly demonstrate that FORMACT consistently outperforms both single-pass and multi-pass baselines.

### Weaknesses
1. In the constructed RichDocBench, the generated documents are evaluated solely by VLM (e.g., gpt-5.2-2025-12-11 used in the paper). To make the benchmark more comprehensive, it is recommended to incorporate sequence-based metrics (e.g., BLEU or Edit Distance) to evaluate structural and semantic fidelity, as well as vision-based metrics (e.g., SSIM or LPIPS) to measure the visual and layout similarity between the generated documents and the ground-truth.

2. It is recommended to provide qualitative visual comparisons between FORMACT and other baselines, as well as visual examples from the constructed datasets. This would make the overall evaluation more rigorous, intuitive, and comprehensive.

3. Although the proposed work is meaningful, the authors do not explicitly commit to open-sourcing the code and datasets. This omission might weaken the potential impact and reproducibility of the work.

4. There is a typo error in Line 372:“See Appendix .”

---

> ### Author Rebuttal · Authors · 2026-03-31
>
> Dear Reviewer 9REF,
>
> Thank you for the constructive feedback, and for recognizing the value of our approach and the proposed benchmarks. We address each concern below.
>
> **Q1: Incorporating Reference-Based Metrics**
>
> Thank you for the suggestion. We respectfully note that rich-format document generation is a one-to-many task, where many valid outputs can differ in section ordering, phrasing, table layout, and visual styling while all being correct and professional. Reference-based metrics penalize these valid variations since they assume a single correct output.
> Prior work has reached a similar conclusion: AutoPage [1] adopts exclusively reference-free evaluation and VLM-as-judge, and Prometheus-Vision [2] argues that traditional reference-based metrics fail to capture the rich context of generated outputs. Following this practice, we adopt VLM rubric grading across four independent dimensions alongside human evaluation. We also explored pairwise win rate, but found that VLM judges exhibit strong positional bias in multi-page document comparisons, and therefore discarded this setting.
>
> **Q2: Qualitative Visual Comparisons**
>
> This is a great suggestion. We agree that visual comparisons would make the evaluation more intuitive. We will include side-by-side rendered page comparisons between FormAct and all baselines, as well as visual examples from RichDocBench and RichDocFuzz, in the revised manuscript.
>
> **Q3: Open-Sourcing Code and Datasets**
>
> We are glad the reviewer raised this point. We are currently undergoing internal compliance review and plan to release RichDocBench, RichDocFuzz, and the FormAct code upon completion of this process.
>
> **Q4: Typo on Line 372**
>
> Thank you for the careful reading. We will carefully double-check the entire manuscript and correct any typos.
>
> We sincerely appreciate the reviewer's thoughtful feedback, which has helped us strengthen the paper. We hope our responses have addressed the concerns raised and welcome any further discussion.
>
> Sincerely,
>
> The Authors
>
> **References**
>
> 1. Ma, Q., Wang, S., Chen, Y., Tang, Y., Yang, Y., Guo, C., ... & Zhang, Z. (2025). Human-Agent Collaborative Paper-to-Page Crafting for Under $0.1. arXiv preprint arXiv:2510.19600.
> 2. Lee, S., Kim, S., Park, S., Kim, G., & Seo, M. (2024, August). Prometheus-vision: Vision-language model as a judge for fine-grained evaluation. In Findings of the Association for Computational Linguistics: ACL 2024 (pp. 11286-11315).

---

> > ### Author Rebuttal · Reviewer_9REF · 2026-04-07
> >
> > The authors' response has successfully resolved my concerns. I will remain supportive of the paper and keep my original positive score.

---

> > > ### Author Response · Authors · 2026-04-08
> > >
> > > Dear Reviewer 9REF,
> > >
> > > Thank you for the positive re-evaluation and for confirming that our responses have resolved your concerns. We are grateful for the constructive and thoughtful feedback throughout this discussion.
> > >
> > > Sincerely,
> > >
> > > The Authors

---

### Decision · Program_Chairs · 2026-04-30

**Decision:**

Accept (regular)

**Comment:**

The paper studies generation of rich format documents from scratch. By operating directly on the HTML source representation it proposes to transform the problem into a document engineering task. The authors then propose an agentic framework to iteratively modify the underlying source HTML, along with a retriever component and a reviewer agent that evaluates and provides feedback on the rendered source.  They propose two benchmarks RichDocBench and RichDocFuzz to further enable development and evaluation in this space.

Overall the results are quite strong and the proposed approach outperforms both single pass generation baselines (Direct Generation and Template-Augmented Generation) as well as multi-pass baselines such as Codex.

All reviewers assigned positive scores and I also support accepting the paper.